# A Comparative Study of Curriculums for Education for Sustainable Development (ESD) in Sweden and Japan

**Ulf Fredriksson [1,\*], Kanako N. Kusanagi [2], Petros Gougoulakis [1], Yaka Matsuda [3] and Yuto Kitamura [4]**

[1] Department of Education, Stockholm University, S-106 91 Stockholm, Sweden; petros.gougoulakis@edu.su.se

[2] Center for Advanced School Education and Evidence-based Research, Graduate School of Education, The University of Tokyo, 7-3-1 Hongo, Bunkyo-ku, Tokyo 113-8654, Japan; kkusanagi@p.u-tokyo.ac.jp

[3] Faculty of Education, Kochi University, Akebonocho, Kochi city, 780-8520, Japan; yaka_m@kochi-u.ac.jp

[4] Graduate School of Education, The University of Tokyo, 7-3-1 Hongo, Bunkyo-ku, Tokyo 113-8654, Japan; yuto@p.u-tokyo.ac.jp

\* Correspondence: ulf.fredriksson@edu.su.se; Tel.: + 46 (0)8-1207-6402

**Abstract:** This study examined the curriculums and implementation of Education for Sustainable Development (ESD) in upper secondary schools in Japan and Sweden and examined and compared the policies and cases of ESD practice. The comparison showed that ESD is present in the national curriculums of both countries, but is emphasized differently. In Sweden, it is more a matter of mentioning ESD as part of the principles that guide education, while in Japan, the integration of ESD into the subject syllabus is emphasized. The schools visited strove to implement ESD in their work. ESD is not an exclusive approach in addition to other school activities, but rather a central part of the schools´ work. Many issues are included under the umbrella of ESD, not only environmental issues. International contacts are one of the most important elements of the work, but this does not exclude local engagement. In fact, the schools have established impressive networks. Project work is an important approach in supporting students' learning about sustainable development. A prerequisite for the successful work of the schools, which all have a certain reputation in the field of ESD, is the dedication of the teachers and the equally dedicated principals who support the work.

**Keywords:** sustainable development; education; comparison; Sweden; Japan; schools

## 1. Introduction

Education has been regarded as an important part of any policy related to sustainable development (SD), and Education for Sustainable Development (ESD) has become an important part of the overall concept of SD. According to UNESCO [1], ESD provides the necessary knowledge, skills, and values as an important approach to preparing "global citizens" as well as nurturing citizens for sustainable societies. ESD has the characteristics of being interdisciplinary, learner-centered, and context-dependent, and empowers learners for societal transformation [2, p. 6].

Sweden and Japan have both played prominent roles in establishing an international agenda for sustainable development. From this perspective, it is of interest to look into the practice of upper secondary schools in Japan and Sweden. The aim of this article is to examine the curriculums and implementation of Education for Sustainable Development (ESD) in Japan and Sweden and then to

analyze how the two schools in Sweden and two schools in Japan work with ESD. By applying a comparative approach, we expect to gain a deeper understanding of innovation in education [3].

The case studies provide insights into the ways the curriculum guidelines on ESD are converted into a didactic approach. The actual implementation of ESD takes place in the school and in the classrooms where teachers interact with and motivate their students. Based on how the general aims of ESD have been elaborated in the national curriculums and earlier research, three areas have been highlighted: (1) how the ESD program is structured in relation to the school's vision; (2) the content of the ESD program; and (3) the networking and cooperation used to realize the ESD program.

The aim of this comparative study is to review the policy and examine four cases of ESD curriculums and implementation in the two countries to provide a more sophisticated understanding of how ESD is organized in different contexts and how vital emerging aspects of education can be examined from an international and comparative perspective. Both Sweden and Japan have a nine-year comprehensive education system. This means that upper-secondary education is not compulsory, but more than 90 percent of students continue their education after completing nine years of comprehensive school [4,5]. Sweden has a decentralized system while Japan has a centralized system. In Sweden, there is a national curriculum developed by the government and the National Agency for Education ("Skolverket" in Swedish), but schools have a large degree of freedom to interpret and implement the curriculum. In Japan, the curriculum is basically developed by the Ministry of Education, Culture, Sports, Science and Technology (MEXT) and the textbooks are authorized by MEXT; this central power therefore regulates schools, allowing limited flexibility. All of the four schools studied in Sweden and Japan are public (state) schools, thus they reflect the respective national policies on ESD.

Rauschmayer and Lessmann [6] argued for the importance of qualitative case study research, which aims to investigate ESD at a micro level, while considering macro data. This project has followed the same approach. Although there are some comparative studies of education in Sweden and Japan [7,8], there have been no previous studies comparing ESD in Japan and Sweden.

## 2. Policy Analysis for Education for Sustainable Development

### 2.1. Education for Sustainable Development (ESD) Curriculum and Policy in Sweden

Education as a means of promoting SD is clearly emphasized in Swedish educational legislation and policy. ESD in the national curriculum is expressed in terms of overall societal goals to achieve a sustainable society [9]. This is indicated in educational legislation [10] and in the curriculum and syllabuses [9]. In the Swedish curriculum for upper secondary schools, SD is mentioned as a goal that can "develop a personal approach to overarching, global environmental issues. Education should illuminate how the functions of society and our ways of living and working can best be adapted to create sustainable development." [11, p. 6]. A conceptual change from "environment" to "sustainable development" coincided with the UN initiative to declare 2005–2014 the Decade of Education for Sustainable Development. Since then, the concept has been exposed to considerable theoretical and ideological deliberations, but there seems to be a widespread consensus regarding its necessity for the quality of life and the maintenance of a welfare society [12].

ESD is not presumed to be the responsibility of only science education, but rather an integrated part of all subjects in the curriculum. This view is reflected in the tradition of the Swedish curriculum, where political, religious, and cultural values are highlighted and expected to be processed by the schools.

Lgy 11 (the curriculum for upper secondary education), gives no distinct definition of sustainable development (SD). It is treated under the section "Fundamental values and tasks" of the Swedish school, and is linked to globalization and internationalization as well as other areas such as democracy, gender, health, identity, cultural diversity, and lifestyle issues. In the same section, four general perspectives are stipulated, intended to permeate the teaching and learning at

school, which in turn corresponds to those that apply to teacher education: *a historical perspective, an international perspective, ethical perspectives* and *environmental perspectives.* The latter should provide students with insights to contribute to preventing harmful environmental effects and developing a personal approach to overarching, global environmental issues [11, p. 6, 13, pp. 192–196]. Education at all levels of the Swedish school system is aimed at promoting students' multifaceted personal development to become active, creative, competent, and responsible individuals and citizens. Emphasis is placed on the students' ability to assess, take a position on complex issues, and to act accordingly.

Nurturing reflecting and action-oriented citizens is stipulated in the school curriculum as one of the main tasks of the education. There are several examples in the governing documents showing teachers how to train students in this ability while working toward sustainability. This goal is promoted through an interdisciplinary approach and through many subject areas that are mentioned in the governing documents [11].

The actualization of sustainability is addressed in all 18 national programs in upper secondary education, but to varying degrees in the constituent subjects. The diploma goals for the Natural Science Program state that the education should help students develop an understanding of how science and changes in society both affect and are affected by each other; in particular, the program highlights the role of science in questions concerning sustainable development. The education should also offer opportunities for students to take part in ethical discussions on the role of science in society, encouraging students to cooperate and stimulating them to see opportunities, try to solve problems, take the initiative, and transform ideas into practical actions [14 p. 227].

With regard to how sustainability issues are presented in the Swedish curriculum for upper secondary education, it should be mentioned that schools and teachers have great autonomy and freedom to determine the forms of education for sustainable development, and together with the students choose to focus on different aspects of it, for example, the local environment, health, lifestyle, consumption, global cooperation, fair trade, and so on.

### 2.2. ESD Curriculum and Policy in Japan

The ESD curriculum in Japan is closely linked to the need to internationalize Japanese society. Education has been repeatedly stressed by various stakeholders since the mid-1980s, as a result of Japan's economic growth and in response to globalization. In particular, the Provisional Council on Education Reform established by the Nakasone Administration in 1984 proposed that international aspects need to be highlighted in educational policy [15] and presented a number of recommendations including a principle emphasizing the individual characteristics of students, efforts to shift from a collective approach to an individual approach in teaching and learning at schools, and a shift to a system of lifelong learning. The report published by the Council is considered an initial response to the changing environments of internationalization and informatization (i.e., the rapid development of information and communication technologies) in Japanese society, which have been continuing to the present.

Since the mid-1990s, Japan has promoted educational reforms, focusing on ways of teaching and learning to respond to the above-stated rapidly changing environments in Japanese society. To prepare students for their future lives in such environments, they are now expected to acquire not only proper cognitive skills, but also non-cognitive skills (or so-called generic or soft skills). For this purpose, the course of study (equivalent to a national curriculum) has focused more on active and participatory learning for students, and has promoted problem-posing and problem-solving education. Thus, the contents of the course of study have been revised to include a wider social and global agenda to deal with the challenges facing humanity today [16].

Based on this background, concepts of sustainable development and sustainable societies were included in the course of study, which was revised in 2008 for preschools/primary schools/junior high schools, and in 2009 for high schools. In the most updated version of the course of study, which was introduced in 2018 for preschools and will be introduced for primary schools/junior high schools/high schools consecutively after 2019, ESD is specifically stated as one of the most

important components. In these coming curriculums, the importance of developing generic skills and competencies is clearly stated.

At the World Summit on Sustainable Development held in Johannesburg, South Africa in 2002, then Japanese Prime Minister Junichiro Koizumi proposed an introduction to the concept of Education for Sustainable Development (ESD), and later, the United Nations adopted the United Nations Decade of Education for Sustainable Development (UN-DESD) for the international community. Since the summit, ESD has been strengthened politically in Japan and this movement has influenced educational fields, mainly through the introduction of ESD at UNESCO Associated Schools across Japan [17].

Japan adopted the National Implementation Plan for the UN-DESD in 2005, revised in 2011, as a national policy initiative. As a first step toward implementing this plan, the Roundtable Meeting at UN-DESD was established within the Japanese government in 2007; it has since become a platform for various experts in their fields to meet and discuss how to promote ESD in Japan. Then, as a next step, the Basic Plan for the Promotion of Education was formulated in 2008 in order to specify the overall policy of the education sector, and ESD was clearly placed as one of the most important principles for Japanese education. At the end of the UN-DESD, Japan hosted the UNESCO World Conference on ESD in Nagoya in 2014. At this concluding conference of the decade, Japan appealed to the domestic and international participants to recognize the importance of involvement by various stakeholders in the implementation of the Global Action Program (GAP), which presents five priority action areas of ESD. As one of the follow-up activities of UN-DESD, the UNESCO-Japan Prize for ESD was launched in 2015 in order to recognize outstanding ESD projects around the world each year. Thus, the diffusion and development of ESD is clearly established as an important policy theme in Japan (This paragraph was extracted from Kitamura [18] and modified and elaborated to fit this article).

In 2015, the Sustainable Development Goals (SDGs) were adopted at the United Nations, and the Japanese National Commission for UNESCO established the Special Committee for the Promotion of SDGs in 2016. As part of the discussion held at this committee, ESD has often been highlighted, and there is a continuous examination of how to promote ESD in schools as well as in communities.

An important policy initiative in Japan to encourage schools and teachers to exercise ESD is the UNESCO Associated School Project Network (ASPnet). The Ministry of Education, Culture, Sports, Science, and Technology (MEXT) has been providing assistance to those Associated Schools. As of February 2018, there are 1033 UNESCO Associated Schools in Japan (This can be compared to Sweden where, according to information from UNESCO, there are 27 UNESCO Associated schools [24]).. The ASPnet is aimed at stimulating the development and extension of international understanding at schools by encouraging national pilot programs linked and coordinated internationally [17].

The UNESCO Associated Schools are not only "elite" schools; they are diverse in terms of geographical location, school size, public/private status, and the academic achievements of their students. At these schools, one of the key elements for promoting ESD is the support from the highly-motivated principals and community members.

There is another policy initiative called the Super Global High School (SGH) program, which was launched in 2014 by MEXT. This program "aims to foster globalized leaders who will be able to play active roles on the international stage through education at high schools that contribute to this mission. Students will achieve goals such as awareness and deep knowledge of social issues, communication ability, and problem-solving skills." (Table 1) [19].

**Table 1.** Comparison of Educatoin for Sustainable Development promotion in Sweden and Japan.

|  | Sweden | Japan |
|---|---|---|
| **National Policy** | Overall societal goal to achieve a sustainable society is expressed in the national curriculum | The need to internationalize Japanese society, to shift from a collectivist approach to an individual approach in 1980s, and to emphasize non-cognitive skills |
| **Curriculum for Upper Secondary Schools** | Developing a personal approach to overarching, global environmental issues. Education should illuminate how the functions of society and our ways of living and working can best be adapted to achieve sustainable development | A wider social and global agenda to deal with the challenges facing humanity today have been included as learning contents, stated in the Course of Study for Upper Secondary School in 2009. |
| **Vehicle behind ESD Promotion** | There is general interest in SD and schools, and teachers have relative autonomy and freedom regarding their pedagogy. Students can focus on different aspects such as the local environment, health, lifestyle, consumption, global cooperation, and fair trade | Schools and teachers tend to combine ESD with other activities such as the ASPnet and SGH programs. Aiming to promote international understanding and foster global leaders |

## 3. Theoretical Perspectives on ESD

As the environment and sustainability are cross-disciplinary issues, the curricular implications of ESD in the formal education system are attracting increasing research interest. ESD is systematically studied and linked to several other general approaches to education, involving a multi-disciplinary approach to science/technology including human rights and global citizenship [20], the environment, the economy, social culture, ethics and policy, and a model termed Sociology/culture, Environment, Economy, Science, Ethics/morality and Policy (SEE-SEP) [21–23]. SEE-SEP is a model that emphasizes the cross-disciplinary character of the complex interrelationship between science and society. It covers six subject areas of socioscientific issues: sociology & culture (S), the environment (E), the economy (E), science (S), ethics & morality (E), and policy (P), connecting them to the three aspects of value, personal experiences, and knowledge.

ESD, like all educational activity, relies mainly on communication, by which students encounter officially sanctioned situations through actual experiences, didactically designed to help them reach specified curricular goals [25]. When teachers design their instruction, they make interpretations and decisions based on the designated curriculum (which is based on the official curriculum), along with other resources. There are differences between the designated and the teacher-intended forms of curriculum that are designed for instruction for specific students at a particular moment and place [26]. The acts of communication unfolding between the teachers and students around the tasks during lessons constitute the enacted curriculum, which may deviate in essential aspects from both the official and the designated curriculums.

Concerning the learning objectives for SD, the national curriculums of Sweden and Japan clearly stipulate the importance of change, which is in line with the theoretical tenets of transformative learning. Transformative learning theory, as developed by Mezirow [27,28], offers an explanation for the change in meaning structures based on Habermas's theory of communicative action. Transformation theory maintains that human learning is grounded in the nature of human communication. For a person to understand the meaning of communication of intentions, values, morals and feelings, critical reflection on assumptions is required. Learning is both instrumental and communicative; the former focuses on task-oriented problem-solving and is oriented toward empirical-analytical discovery [29]. Communicative learning is about understanding the meaning of what others "communicate concerning values, ideals, feelings, moral decisions, and such concepts as freedom, justice, love, labor, autonomy, commitment and democracy" [30, p. 8, cited in 29, p. 5]. Transformative learning occurs when these two domains of learning involve "reflective assessment of premises" and revision of meaning structures and schemes. Meaning structures act as culturally defined frames of reference and comprise meaning schemes and perspectives.

Meaning schemes are "made up of specific knowledge, beliefs, value judgments and feelings that constitute interpretations of experience," while meaning perspectives are general frames of reference.

## 4. Case Studies

### 4.1. Methods

The aim of this section is to present a comparison of cases of how two schools in Japan and two schools in Sweden have interpreted the ESD curriculum and implemented it. The participating schools were selected based on their emphasis on the ESD curriculum as well as their reputations as "good examples." In addition to these criteria, the schools were also willing to participate and receive researchers from Japan and Sweden. The five authors—all of which are comparative educationalists—formed a research team and all visited the school sites. Their backgrounds are diverse: one has extensive experience with ESD policy formulation, having served as an advisor for UNESCO and JICA; one is a former school teacher who has expertise in school curriculums; one is an expert in lifelong learning; one is a researcher working in the fields of both Swedish and Japanese educational institutions; and one is a former consultant who has been working with teacher transformation in various countries. The rich experience of these authors has contributed to the provision of the multiple perspectives needed for the ESD study.

The research team's visits to Japanese schools took place in October 2017, and those to the Swedish schools in November 2017. In the case of the Japanese schools, the Japanese authors worked with the schools and arranged our visits with the ESD coordinators. In the case of Sweden, the authors received recommendations from the local education administrations and then arranged visits directly with the schools. At each school visit, the research team organized focus group interviews with the principal, key teachers, and selected students. The interviews were guided to focus on questions about the history of the school, and the goals, curriculum, implementation, and evaluation of the ESD activities.

Additionally, more specific questions were asked regarding the ESD curriculum and implementation such as the advantages/disadvantages of including ESD into the curriculum, the kinds of methods/pedagogy used to include ESD, the kinds of skills/abilities students acquire through ESD and how this is evaluated, and future perspectives on ESD. The team walked around the school facilities and observed some lessons. Informed consent was gained from all the interviewees. The researchers also looked at the school facilities in order to gain an understanding of the general ambience of the schools, and were provided with relevant material on ESD activities in the schools. Information about the schools was also collected from the schools' websites and ESD-related reports. While the sample data size was limited, these cases provide insights into the benefits and challenges of implementing ESD at the upper secondary school level.

### 4.2. Overview of School A in Japan

School A is a university-affiliated school located 60 kilometers from Tokyo. The school is distinctive in that it offers integrated courses (*sougou gakka*): both general courses and technical (vocational) courses. According to MEXT, *sougou gakka* have the characteristics of (1) offering a wide variety of elective subjects for students to choose from, which enables them to pursue their interests and actively participate in learning; and (2) specializing in career education [31]. Students choose from one of four cluster subjects: (1) Biological Resource and Environmental Science; (2) Engineering System and Information Science; (3) Life/Human Science; and (4) Human Sociology/Communication. The ESD concepts permeate all four majors.

The strength of School A is its curriculum flexibility in the form of *sougou gakka*. Since students are generally attracted by the unique programs offered by the school, the school is not too concerned with university entrance exam preparation. Due to this advantage, School A is able to engage in ESD as a whole-school initiative and can spend more time on various programs. School A sets the following three targets:

(1). To approach building a sustainable society from diverse subjects unique to an integrated high school;

(2). To promote learning from the viewpoint of ESD and further promote inter-subject collaboration and learning; and

(3). To provide the opportunity to exchange research with other high school students both in Japan and abroad, and experience learning from an international perspective

The goals of School A clearly resonate with the principles of ESD. The school aims to prepare students to adjust to changing societies and technologies as citizens for a sustainable society, and nurture them with competencies and skills for lifelong learning by offering integrated courses. By experiencing programs with a global perspective, students are expected to develop (1) core academic skills, (2) a global perspective, and (3) flexible thinking.

### 4.2.1. ESD Program

School A offers several programs relevant to ESD. The school is a member of the UNESCO Associated Schools Project Network (ASPnet), is designated as a Super Global High School (SGH), and is certified as an International Baccalaureate Diploma Program School (IB). Even before becoming a UNESCO School, School A had already engaged in ESD-like activities because the school's mission was already in line with the ESD principles. The integrated courses offer a practice-oriented approach and multi-disciplinary and global perspectives in learning what ESD promotes. After the school became an official UNESCO School, these activities were upgraded and organized as ESD activities. As discussed, this school revised its school goals to include the ESD principles in 2011. After this, it became a UNESCO School in 2011, and was designated as an SGH in 2014. By being a UNESCO School and receiving funding for SGH, the school was able to obtain more resources in terms of networking and funding, which contributed to the further development of ESD. Thus, rather than separate programs, these programs should be considered as multi-layered and having a synergetic effect.

In the ESD/SGH program, all first-year students have a lesson called "Global Life" as part of the Home Economics curriculum. This program aims to engage students in local/global issues with four lenses: (1) integrated, (2) contextual, (3) critical, and (4) transformative, which are necessary for them to become global citizens. In the second year, students spend 20–30 hours engaging in group research to investigate a local issue linked to global life, focusing particularly on Association of Southeast Asian Nations　(ASEAN) issues. Student representatives also participate in fieldwork in Indonesia and engage in ESD-related activities collaboratively with Indonesian high school students. One example is consuming local agricultural products. Graduation Research is a mandatory course for third year students, where students can choose any topic based on what they have learned at school (many of them choose agricultural issues) or their future plans at university. By engaging in these activities, students develop various skills including how to conduct research, how to communicate with others, and how to present their research. Since this school has an affiliation with a university, students can visit the university to learn from the cutting-edge research of professors and graduate school students. The school also has partnerships with schools and universities abroad as well as Non-Governmental Organizatoins (NGOs) and private companies and makes use of these resources to provide various opportunities for students.

### 4.2.2. Networking and Cooperation

School A has developed various networks to provide students with first-hand experience of working with foreign students. Every year, the school participates in a symposium in which students from ASEAN countries share their ideas to create a sustainable society. In particular, graduation research provides a good opportunity for students to explore their own interests, make their own hypotheses, conduct research and experiments, and reflect and present their findings. One of the students wrote in an essay that the image of a global citizen has changed from being able to speak English to being aware of global issues, being able to connect with other people, and

being able to carry out one's own actions. At the same time, student surveys conducted by the school indicated that students also faced challenges working with people from different backgrounds, which did not always result in positive experiences.

On the other hand, while this school aims at a multi-disciplinary approach across subjects, there are challenges in consistently carrying out ESD across different subjects. Some teachers do not understand the concept of ESD and there are limitations, since there are various degrees of understanding and commitment among teachers to ESD programs. In addition, since this school has ESD, "Super Global High School" (SGH), International Baccalaureate (IB), and other programs, there is always the challenge of balancing time and resources for these programs.

*4.3. Overview of School B in Japan*

School B is an upper secondary school in Gifu Prefecture, located in central Japan. This school has 900 students and 60 teachers. It has a high reputation thanks to its academic excellence, and sends students to well-known universities.

School B has also been a member of the UNESCO Associated Schools Project Network (ASPnet) since 2017, and was designated as a "Super Global High School" (SGH) between 2014 and 2019. Of the 60 teachers, 36 or so are engaged in the SGH program, which was implemented to develop "global leaders." The goals of the program are set as follows:

(1). Students who play active roles as leaders in the international world;
(2). Students who have acquired a high level of communication skills in a foreign language;
(3). Students who can discover a problem and find ways to solve them, and work together with others to resolve the problems;
(4). Students who have acquired methods of thinking about and expressing topics critically and logically as well as from varied and comprehensive perspectives; and
(5). Students who can appreciate different cultures and values and have a wide range of education concerning traditional Japanese history and culture.

4.3.1. ESD Program

There are five research themes that students can choose from:

1)  International Development: Students study the current problems of development that Asian countries are facing and the trends of their international contribution policies;
2)  International Business: Students study local companies that expand their management strategies globally and the possibilities relating to their corporate social responsibility;
3)  Environmental Energy: Students study international contributions in the area of environmental energy, aiming at combining both the humanities and science courses;
4)  International Medical Treatment: Students study medical problems, specifically concerning human immunodeficiency virus (HIV), focusing on international medical welfare and researching the relationships between Japan and the international world; and
5)  Comparative Education: Students study the current national trends through educational policies and practices encouraged globally.

Lectures and lessons in relation to a sustainable society were provided to encourage students to engage in problem-solving. For example, the first-year students made a presentation based on the field study in the following process (Figure 1).

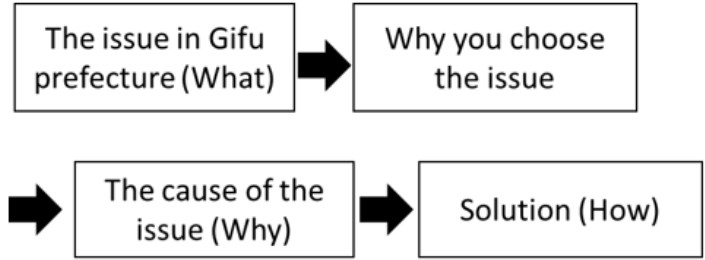

**Figure 1.** Presentation of the field study.

Another example is the second-year students attending a workshop called a "Problem-solving workshop". A guest lecturer came from a university and gave a lecture about "Sustainable Community Development," and the students learned about "Sustainable Community Development" through discussions.

### 4.3.2. Networking and Cooperation

School B cooperates with people and organizations both locally and globally. Students participate in field studies in Cambodia and Vietnam, and they visit both Japanese and local companies. They also visit Cambodian and Vietnamese high schools and talk to local students. Furthermore, the students visit different laboratories in Japanese universities to deepen their knowledge of a research theme. In addition, students participate in social activities such as a "Cleaning Activity." In the cleaning activity, students visit the homes of elderly people and people with disabilities in the neighborhood and support these people by cleaning their homes and socializing with them. This activity is a good opportunity for both the students and the local residents to meet and talk to each other.

### 4.4. Overview of School C in Sweden

School C is an upper secondary school located in central Stockholm, founded in 2003; it is based on a vision of a group of teachers committed to raising global citizens. It gained support from the city of Stockholm to become a profiled municipal school. The school has about 700 students and 60 teachers. It offers four higher education preparatory programs: Art and Design, Natural Science, Social Science, and Business and Administration. In addition to these programs, special courses are offered to newly arrived immigrants.

The school emphasizes the principals of ESD, and the goals for the SDGs are stated in the school brochure. All programs focus on ESD, and a multidisciplinary approach is used. Through the school's motto, "Caring, trust, respect, and responsibility", the vision is to educate students to gain:

- a creative, critical and analytical way of thinking;
- a better understanding of themselves and their potential for personal development;
- a realization of self-development together with and through other people;
- the strength to implement their own visions and to consider the impact of their actions;
- insights into global issues, including both problems and possibilities for change; and
- an understanding of the fact that our future must be built on sustainable development.

Many teachers at the school actively applied for positions at the school as they shared the values expressed through the school profile. A large proportion of the teachers are actively involved and interested in developing their teaching methods. Professional development takes place in different ways for the teachers during a "typical" working week.

4.4.1. ESD Program

ESD is an integral part of all programs and activities offered by the school, and the global agenda is reflected in many of the school's activities. Learning for sustainable development includes a global perspective, student democracy, access to the world, and interdisciplinary projects. A variety of pedagogical methods are used including both formative and collaborative learning. The students are regarded as active contributors to the work of the school.

All subjects are approached from a multidisciplinary perspective. Much of the teaching is based on project work, and there is a flexible schedule that changes from week to week. Use of and access to computers is a prerequisite for the flexible working methods of the school. Each student has their own laptop, and the school actively uses a digital platform. A culture of sharing is encouraged. Learning through communication is seen as an objective. Google Drive is frequently used and the Internet is seen as a means of gaining access to the world.

The students spend about half of their time working on projects. The students participate in interdisciplinary projects during four periods every year. They work in project groups and have a project tutor. The results of the project work are examined. Examples of projects are "Out in the Forest", "Glocal Democracy", "Migration", "Model UN Roleplaying", "Global Health", "The Others", "Sun, Wind and Water", "The Silent Sea", and "Food for Your Mood".

Practical learning through field studies, excursions, study visits, meetings with different people, and work with real issues is seen as an important part of the school's activities. Field studies are organized as part of the studies in the third year of upper secondary education.

The students are given many opportunities and are encouraged to actively take part in the school management. School democracy is regarded as an integral part of the school. There is an active student council with various subsidiary groups/student clubs. There are student meetings in class where they meet with a team of teachers, and sometimes, school meetings with principals/teachers from all teams. There is a general commitment to social activities.

4.4.2. Networking and Cooperation

The school actively cooperates with several organizations and institutions. Part of the cooperation is internal at the school, where there are different student organizations actively working with different issues and are provided with space within the school.

There is cooperation with organizations and institutions outside the school such as Amnesty International, Plan Sweden, Social Entrepreneurship Forum, and World Wildlife Fund (WWF). The school is one of the model schools of the WWF. Together with Plan Sweden, field studies have been organized in various countries for students.

Field studies form an important part of the practical learning organized at the school. They are organized as three-week experiences. In total, the school has organized about 30 field studies in India, Malawi, Nicaragua, Uganda, and Bangladesh. There are also other types of international contact with schools in other countries through the establishment of exchange programs. Such programs exist with schools in Ukraine and India.

Contacts are established with the city authorities in Stockholm, and the students become involved in city planning. There is also cooperation with the Stockholm Resilience Center (SRC) at Stockholm University and the Royal Technical College in Stockholm. Students can also choose a course on sustainability, "Världens Eko", which is organized at Stockholm University/Resilience Center.

*4.5. Overview of School D in Sweden*

School D is an upper secondary school in the county of Östergötland in southeastern Sweden, about 200 km southwest of Stockholm. It is one of the oldest upper secondary schools in Sweden, which is reflected in the school motto: "Tradition and development". It is a municipal school owned by the city where it is located. The school offers higher education preparatory programs including programs on the economy, trade and administration, and humanistic and natural science. All

programs last three years. In addition to these programs, the International Baccalaureate (IB) program is offered. There is an active student council at the school, which represents the students in different contexts and organizes activities for them. A number of subcommittees comprise one part of the student council. In addition to the student council and its subcommittees, there are also many societies at the school, some of which are long-established.

### 4.5.1. ESD Program

Student project work is organized from a multidisciplinary approach. Different types of projects are organized. One project with a science focus is organized during the first year in the study programs. Later on, the students are involved in projects they choose for themselves.

A science project that has been organized on several occasions is "Science and our sustainable world: An interdisciplinary project". This is an interdisciplinary project based on the question: "Is it possible to detect traces of the Cold War in our immediate environment?" To answer that question, the classes take a core sample from a peat bog. With the support of the lab at the nearby university, radioactive fallout including Cs-137 has been detected in a core sample.

The project is expected to have synergy effects such as the following:

- Providing an outdoor lesson to show that physics and other school subjects exist outside the classroom;
- Creating a common reference for the future, when discussing radioactivity and other abstract subjects;
- Relevance from measuring something real;
- Demonstrating that science is a necessary part of human life on Earth;
- The work is performed in collaboration with a university, giving the students an insight into what a scientist might work with;
- Pollution is placed in a context encompassing biology, history, chemistry, etc.; and
- The students write their first draft of a scientific report.

### 4.5.2. Networking and Cooperation

The general aims expressed by the school in an action plan for internationalization are to increase the students' potential to reach their learning goals and increase their quality of education through contact and meetings with students and teachers from other countries. Young people need to develop their identities by meeting other people, where they can understand their situations. Such meetings also help them improve their skills in and knowledge of foreign languages.

The school has been active in seeking cooperation with other schools working with environmental issues. This has been part of the international network for ecological schools [ECO schools]. Together with four other schools in the Netherlands, France, Hungary, and Greece, the school has applied for and received Erasmus+ funding for student and teacher exchanges in 2017–2020. The main purpose of the project is to create, together with the other schools, an application for long-term work related to sustainability. As part of the work with the ECCO schools, the school has also worked with the Green Flag project in Sweden [32].

The school has also actively sought cooperation with other schools through exchange programs. Such programs exist with schools in Germany, France, the U.K., Finland, and Spain. These programs have included regular study visits to these schools. There is also ongoing work to expand international contacts with schools in the U.S. and China.

The existence of the International Baccalaureate (IB) program at the school and the fact that 30 different languages are spoken as mother tongues by the school's students adds important dimensions to the global perspectives of the school.

## 5. Discussion and Conclusions

### 5.1. Main Findings

The case studies provide insights into the ways in which the ESD curriculum is interpreted by teachers and how lessons are designed and implemented at the school level in Japan and Sweden. As mentioned in the introduction, three themes are discussed in the case studies: (1) How ESD is structured in relation to the school vision; (2) the content of the ESD programs; and (3) networking and cooperation to realize the ESD programs (Table 2).

**Table 2.** Comparison of school mission at the four schools.

| | School Vision |
|---|---|
| **School A (Japan)** | The school's mission is to offer integrated courses, prepare students to adjust to changing societies and technologies as citizens for a sustainable society, and nurture them with competencies and skills for lifelong learning. |
| **School B (Japan)** | The school's educational goal is to nurture healthy students who have intelligence, virtue, and an affinity for their fellow man. The school hopes that students will devote themselves to peace and promote understanding between people of the world. |
| **School C (Sweden)** | The school develops pedagogical methods to increase students' commitment to global development issues, and provides opportunities to participate in and influence social development. |
| **School D (Sweden)** | The school has the task of transferring values, conveying knowledge, preparing the students for work, and to be responsible citizens. The school conveys knowledge that constitutes the common frame of reference in society based on fundamental democratic values and the human rights we are subject to. |

In order to nurture global citizens, all the schools have conducted projects relevant to engaging in local and international issues. Local issues seem to be more emphasized in schools A, B, and D. School C is also engaged in local issues, but global issues seem to overshadow local issues at the school to some extent. The surrounding community can be present through direct engagement in issues of local interest, but also through contact with local NGOs and institutions. Such contact also includes contact with universities, both in the neighborhood of the schools and elsewhere. Contact with the private sector seems to be less prevalent than contact with different types of organizations. All of the schools have developed some type of network to support their work.

ESD is not only about environmental education. All the schools place ESD in a larger context, but, for example, school D seems to emphasize traditional environmental issues slightly more than the other schools. Many of the learning activities are motivated by arguments related to the need to provide young persons with the tools for social transformation. Active learning is emphasized and the intention is to achieve this through project work, fieldwork, and integrated/cross-curricular studies. There is a degree of flexibility in the way in which the schools organize their education. At School B, the students' learning process could be seen as a part of the SGH program. In the first year, they study in groups at the beginning of the academic year, studying and thinking about the local community, and in the second year, they study individually and think globally. Furthermore, school D has plans to further develop the project work.

All schools were obviously aware of the global agenda for SD and the role of ESD within it, but this was differently expressed. School C seemed to be the school that more directly discussed SD and ESD. References to Agenda 2030 were frequent in all of the material from the school, for example, in the official brochure presenting the school and on the school website. Additionally, at School A, many references were made to SD. At School B, students do not specifically learn about SD; however, they have many opportunities to learn and think about SD. It could be said that the

challenges and educational practices relating to SD at this school are ESD. Generally, ESD is very much seen as being linked to internationalization and the ambitions of the schools to build international networks.

All of the schools are committed to social activities, which are supported by the flexibility they have in organizing activities at the schools. In addition to activities during lessons, there are also other activities. It is relevant to discuss a whole school approach. Schools C and D have active student councils/students unions that are involved in the schools' decision-making. In all of the schools, there are various clubs and associations for students that organize social activities connected with issues related to SD.

In addition to the three areas of particular interest, it can also be concluded that strong supporting structures are present in the schools. There are dedicated teachers at the school and they are given opportunities to act. There is active support from the principal and, when relevant, from the local school board. In many cases, this is not just a matter of support, but the principal may be seen as one of the dedicated individuals who actively work with ESD. All the schools have physical facilities including classrooms and other premises, which make it possible to work in an active way with SD issues. The schools have supporting networks with external persons and other resources that can be mobilized to support ESD for time-limited initiatives, but also for long-term support. To some extent, the schools also have clear goals for ESD teaching and a clear but basic definition of ESD, but this varies to some extent between the schools.

One more issue of interest is that the schools have a more or less elaborate vision of teaching and learning, and ESD is an integral part of that vision. Generally, the schools are actively involved in many different kinds of activities and seem to have a history of being innovative. The leadership style of the principal, as above-mentioned, seems to be a crucial factor. ESD is an integrated part of the schools' culture, which is most obvious in School C. The teachers' professional development has to some extent been part of the work with ESD, and teacher collaboration is an important element that makes ESD in itself sustainable at the schools. External policies and linkages relevant for ESD are crucial at all of the schools.

*5.2. Comparison: Differences and Similarities*

At the policy level, both Japan and Sweden are committed to promoting SD and ESD internationally. This is reflected in the national interest to support ESD. Concepts of SD and ESD are integrated into the national curriculums and syllabuses of both countries. One possible observation is that the Swedish curriculum to a larger extent emphasizes ESD as part of the general objectives, while in Japan, the emphasis is more on issues in the syllabus. There are a number of national initiatives within the Japanese education system to internationalize education. These initiatives have a direct link to and importance with regard to the implementation of ESD in the schools. The Swedish education system is more decentralized, which means that national initiatives such as those in Japan, are not present in the same way. On the other hand, the schools do have a lot of freedom to decide how to implement the curriculum, and schools are even encouraged to actively create their own profiles, which is clearly illustrated by School C in this study.

The extent to which the schools can elaborate different ways of approaching ESD is also dependent on more general structures in the systems. In the Japanese education system, there is fierce competition among students to enter better-ranked universities. This limits, to some extent, the possibilities for individual schools to diverge too much from other schools during at least the final year of upper secondary education. Although there is a similar issue in Swedish schools, there is less emphasis on the entrance exam compared to the Japanese system.

When the four schools are compared, some obvious similarities can be observed. All the schools have adopted an expanded approach toward ESD, but there are differences between the schools in terms of what seems to be emphasized. Project work is an important instrument for involving students in work around ESD. Project work includes, to a large degree, an interdisciplinary approach where the students are actively encouraged to use their knowledge from

several subjects. Support from authorities/organizations is important for the schools, and supportive school management and enthusiastic teachers were found at all of the schools.

It is also possible to observe some differences. The degree of centralization/decentralization of the school system has implications for the teachers' pedagogical space of action, and thus flexibility in designing ESD activities. In a decentralized system like that of Sweden, schools have more opportunities to find their own ways of implementing ESD. In a centralized system like the Japanese system, there are, on the other hand, more central initiatives that can be seen as support for the schools when they try to implement different parts of the curriculum. Which of these two models, decentralized or centralized, is most effective with regard to SD goal achievement could be the subject of a thorough comparative study in which more school systems are considered.

A global perspective includes active work and contact with school projects in other countries, but the geographical focus of the schools shift. Schools A and B have both developed contacts with schools and projects in different Asian countries., which include both field studies in those countries and the active exchange of students. School C has, over the years, developed contacts with many schools and projects in many parts of the world including Africa, Asia, and Latin America. The school organizes field trips to projects in these countries on a regular basis. School D has a more European approach and, through various European cooperation projects, has built up a network of contacts that focuses on environmental questions. The school has exchanges within these projects, but also exchanges with individual schools with a focus on language learning.

An interesting observation is that Schools A and B actively referred to UNESCO, and are active UNESCO Associated Schools. At Schools C and D, there were hardly any references to UNESCO despite the active references to Agenda 2030 at School C. When they talked about Agenda 2030, it was in relation to the UN as a whole. At School D, which is actively involved in different European networks, the EU seemed to be the international organization most often referred to. UNESCO Associated Schools were not mentioned by Schools C and D.

## 6. Conclusions

The purpose of this article is to look at both how ESD is represented in the curriculums for upper secondary education in Sweden and Japan, and how ESD is actually implemented in upper secondary schools in the respective countries. While the implementation of ESD is encouraged in many countries, there is often trial and error in its actual implementation at the school level. By providing a comparison of the ESD policies, curriculums, and implementation in Japan and Sweden, this study could help people foster better understanding of ESD against different contextual backgrounds. The general comparison between the curriculums shows that ESD is present in the national curriculums of both countries, but is emphasized partly in different ways. In Sweden, it is more a matter of mentioning ESD as a part of the general principles that should guide education, while in Japan, the integration of ESD into the syllabuses in the different subjects is emphasized. This could reflect the general structures of the education systems of the two countries, with the Swedish system being more decentralized than the Japanese. It would certainly be of interest to follow up this study with more detailed studies of the curriculums and the different subject syllabuses.

Despite the fact that the two countries are situated in different parts of the world with different traditions and cultures, all four schools witnessed positive impacts and changes in the students from engaging in the ESD activities. Although the number of cases in this study was limited, we were able to identify some differences and more similarities among the cases. Compared to regular schools, all four case schools, but especially the Japanese schools, emphasized "global perspectives". By promoting ESD, the younger generation are becoming more aware of SD, which can contribute to the realization of SDGs.

For these schools, ESD is not an exclusive approach in addition to other school activities, but is a central part of the schools´ work. Many different issues are included under the umbrella of ESD, and not only environmental issues. Internationalization and international contacts seem to be one of the most important elements of their work. Internationalization does not exclude local

engagement. To promote both international and local contacts, the schools have all established impressive networks. Project work seems to be an important approach in helping students learn about SD and develop their values and critical reflections. To integrate project work in the general organization of the schools, there is a need to be flexible from a pedagogical point of view. An important condition, which may be a prerequisite for the successful work of the schools, which all have a certain reputation in the field of ESD, is dedicated teachers and equally dedicated principals who actively support the work.

Only a very small number of cases have been presented in this study, but since they all have a reputation in the field of ESD, they may give some hints about how schools can work to implement ESD in their education. It would be relevant to expand this type of study to a larger number of schools, both to discover a greater variety of practices and to gain a better idea of the frequency of existing practices in Sweden and Japan.

**Author Contributions:** All authors contributed equally to this paper. All authors have read and agreed to the published version of the manuscript.

**Funding:** Funding from The Scandinavia-Japan Sasakawa Foundation made it possible to visit Japan in October 2017, and to Sweden in November 2017. The research was supported by JSPS KAKENHI, Grant Number 17K18612.

**Conflicts of Interest:** The authors declare no conflicts of interest.

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
