# Peer review of "A Comparative Study of Curriculums for Education for Sustainable Development (ESD) in Sweden and Japan"

_sustainability, doi:10.3390/su12031123_

Round 1

Reviewer 1 Report

Policy analysis

Theoretical perspecitives

Case studies

Methods

Overview of each school (4 chapters)

ESD in each school

Network and cooperation

Discussions and conclusion

Main findings T

Table – comparative on school visions

Comparisons – differences 582  and similarities 575

Language and presentation

The English is not perfect but will need a good final proof read to check the

Much of the data was collected on-site with all five researchers visiting each school (is this right?). How were the data collected and then analysed? How was the structure  of school visits organised? Who accompanied the visitors? Who were their contacts in the schools.

Interview/visit/school? Was there a common checklist used in each school e.g. as in some OECD studies?

The title includes the term „pedagogy“ but the actual text is virtually devoid of a discussion of pedagogy and no defintions could be foun

Author Response

A professional proofreader reviewed the paper. We added more information in 4.1 Methods. We have eliminated "pedagogy" from the title, as suggested by the reviewer.

Reviewer 2 Report

Overall, this is a great study which will be of interest to many researchers. A few suggestions:

The introduction and parts of the paper are written in the future or present tense, as if the study has not yet been conducted. Since the study has been conducted, change to past tense. There are many references to UN sustainable development goals for the years leading up to 2015. A discussion of how or whether these have been met in the schools studied would help, or of UN sustainable development goals in the period after 2015 (considering the study took place in 2017). The method for research is described as 'case study.' The authors could be specific on the type of case study (in this case, maybe it is a multiple case study). The method section also mentions the authors. A very brief description of their knowledge or experiences with ESD might help. A brief description of the permissions needed to conduct interviews/discussions with participants would be very helpful. The authors mention that they held 'discussions' with teachers, school leaders, and students. It would be helpful to see the 'discussion or interview questions' they had for the participants. For example, what questions did they ask, or what was the content of the discussions? The conclusion can be made stronger by making a case for ESD. 

Author Response

Thank you for your constructive feedback.

We have accommodated to change of tenses. Methods section was amended to reflect reviewers' comments. The conclusion was revised to provide additional information.

Reviewer 3 Report

The topic is interesting and up-to-date, however, there are things that must be corrected before the article considered for publication:

The authors compare two countries, that differ not only geographically, but their social and cultural difficulties could impact the research results, hence, the authors should explain how they standardised the results. It s necessary to explain how the comparison of Sweden and Japan could contribute to the scientific literature and what is the value-added of that comparison. The authors present the aim of the study, however, they do not formulate the problem. Because of that, it is not clear based on what the aim is formulated. In section 2.1 and 2.2 ESD curriculum and policy in Japan and Sweden are presented. However, the presentation is not enough. The similarities and differences should be identified and discussed. Section 4.1 is entitled "Methods", but I failed to find any methods' description. The authors presented only the steps of the research. Moreover, the presentation of the steps is vague; the authors ought to present the algorithm of the research more thoroughly. I suggest visualising it.

Author Response

Thank you for your constructive feedback.

We have revised the introduction and conclusion to clarify the aim and arguments. 4.1 Methods section was amended to provide more information regarding how we collected data and analyzed it. A table was added in Section 2.2 to visualize similarities and differences.

Round 2

Reviewer 1 Report

The alterations are in order and most of the suggestions have been dealt with. It was an interesting paper and good luck to the authors.